# OpenReview forum: "Jailbreaking Language Models at Scale via Persona Modulation"
_ICLR.cc/2024/Conference — Submitted to ICLR 2024_

### Official Review · Reviewer_quGs · 2023-10-27

**Soundness:** 2 fair
**Presentation:** 3 good
**Contribution:** 2 fair
**Rating:** 5
**Confidence:** 3

**Summary:**

This paper proposes a new method to generate attack prompts for LLMs. The general idea is to have several stages of generation, where an unsafe category is combined with a persona to generate the attack prompts. The paper found that the method can lead to high attack success rates against several LLMs.

**Strengths:**

1. the presented method is intuitive and clearly presented.
2. the method is tested on several different models to show its effectiveness
3. the method is also simple to implement, and it addresses an important problem.

**Weaknesses:**

1. the biggest weakness is the evaluation method used in the paper. The authors claim that they use an LLM to evaluate the attack success rate through few-shot prompting. However, it's hard to trust such an automatic evaluation method. The authors should also conduct human evaluation for the method.

**Questions:**

Have you conducted human evaluation? It shouldn't be too hard since everything is in English.

---

> ### Author Response · Authors · 2023-11-16
>
> Thank you for your feedback. We have written a [general response](https://openreview.net/forum?id=gYa9R2Pmp8&noteId=2gC4CNvHds) that can also help clarify our work.
>
> > human evaluation for the method is required
>
> **We conduct human evaluation of our method**, we are sorry if it was not clear enough. We manually labeled 300 random generations and used them as gold-standard to measure the effectiveness of our automated classifier. Please, see Appendix A for all details on the evaluation. Let us know if you have any further questions, we are happy to elaborate more.
>
> If this was the main reason for rejecting the paper, we would really appreciate if you could revisit your decision. We are happy to discuss any other concerns you have during the rebuttal.

---

> > ### Author Response · Authors · 2023-11-20
> >
> > As the rebuttal phase deadline approaches, we wanted to check if you have any concerns that our comments and updated version have not addressed. We want to make sure all major concerns are fixed for the meta-review.

---

### Official Review · Reviewer_2ytU · 2023-10-28

**Soundness:** 2 fair
**Presentation:** 2 fair
**Contribution:** 2 fair
**Rating:** 3
**Confidence:** 4

**Summary:**

This work explores jailbreaking LLMs by adding particular persona-related system prompt. Particularly, they design an automated (or semi-automated) way to search the system prompt based on the attacking field. The empirical results show that personalized system prompt could effectively stimulate the LLMs to generate harmful outputs.

**Strengths:**

1. The proposal to induce an assertive persona within a Large Language Model (LLM) is well-founded. Consequently, this paper merits increased attention concerning the use of system prompts in real-world applications.

2. The empirical results regarding Persona-modulated Human Response (HR) highlight the vulnerabilities associated with modifying system prompts, which currently remain accessible without restrictions.

**Weaknesses:**

1. It is advisable to incorporate additional baseline comparisons within the empirical results section. For instance, it would be valuable to assess the performance when transitioning from a persona-modulated prompt within the system prompt to one within the user prompt.

2. The methodological exposition, particularly in relation to the automated procedures, would benefit from greater elaboration. For instance, it would be helpful to provide specifics on the prompting template utilized for the automatic generation of persona-modulated prompts by GPT-4.

3. The outcomes stemming from the semi-automated red-teaming pipeline warrant quantification. This quantification is essential for a comprehensive evaluation of the trade-off between efficacy and efficiency in the context of the study.

**Questions:**

1. what is the performance of moving the persona-modulated prompt from system prompt to user prompt?
2. what is the prompting template for GPT4 to automatically generate the persona-modulated prompts？
3. What are the quantified results of semi-automated approach. Are there any illustration about the trade-off between its effectiveness and efficiency?

**Details Of Ethics Concerns:**

The paper is about jailbreaking LLMs, which may raise ethics concerns about the safety of LLMs.

---

> ### Author Response · Authors · 2023-11-16
>
> Thank you for your feedback. We have written a [general response](https://openreview.net/forum?id=gYa9R2Pmp8&noteId=2gC4CNvHds) that can also help clarify some of the issues you raised. We cite and address the main concerns in your review below.
>
> > assess performance when transitioning persona-modulated prompt from the system prompt to user prompt.
>
> We do not directly ablate this, but **Claude 2 and Vicuna take the persona-modulation prompt in the user prompt** since they do not support system prompts. This indirectly serves as evidence that our method does not strongly rely on the type of prompt used. We think re-running the experiments in GPT-4 user prompt is very costly and would provide little value since our main goal is to show this attack works rather than optimizing its performance. Please, if you think otherwise, let us know and we will be happy to discuss or consider further experiments.
>
> > provide specific prompt templates
>
> We discussed this in the general response. **To prevent misuse, we are sharing prompts and details with trusted researchers** upon request. We included a “broader impact” paragraph discussing this decision in the updated PDF.
>
> > The outcomes stemming from the semi-automated red-teaming pipeline warrant quantification.
>
> Thanks for bringing this up. This is not easy since quantification may strongly depend on the expertise of the human and how long they iterate on an attack. In our internal experiments, the authors could achieve close to 100% harmful completion rates with modest efforts. We illustrate the tradeoffs between the attack modalities in Figure 5 in Appendix B. It includes some estimates of the time required by each attack modality based on our experience. In Appendix E you can find specific harmful generations for prompts that are used as safety examples in the GPT-4 system card. All in all, we believe this section is meant to be a qualitative contribution rather than a quantitative one. We just want to motivate that issues with automations can be easily fixed by having a human in the loop. Please, let us know if there are any specific metrics you think could help improve this point.
>
> > Ethics concern: Potentially harmful insights, methodologies and applications
>
> Thanks for bringing this up. This is a well-known dilemma in security research. It is well-established in computer security that properly disclosing vulnerabilities is better than keeping them private. In general, we think this work does not enable any new dangerous behavior and we kept details private to prevent misuse by a wider audience. **We included a new “Broader impact” paragraph** in the updated version of the paper addressing this point.

---

> > ### Comment · Reviewer_2ytU · 2023-11-17
> >
> > Thanks for the reply. It would be appreciated if the authors could answer the following questions.
> >
> > Based on the Persona-modulated Prompt in Table 1, the author uses [System] and [User] to differentiate the persona-modulated Prompt and the question specific prompt. Will the [system] part (persona-modulated prompt) be placed in the system prompt or the user prompt when doing inference. If it is placed in the user prompt, what is the format to combine the persona-modulated prompt and the specific question prompt?
> >
> > Regarding the responses of system prompt, the author mentioned that Claude 2 and Vicuna take the persona-modulation prompt in the user prompt since they do not support system prompts. To my knowledge, the API-based models like GPT4 and Claude2 do provide interface for system prompt (https://platform.openai.com/docs/api-reference/chat/create) where the user pass values of “role” to determine what type of prompt it is. The default system prompt is important for human aligned chat models. For opensource models like vicuna, they also have official chat templates (check the chat template of vllm https://github.com/vllm-project/vllm) and it is well-known that chat models are sensitive to system prompts. That’s why I raise the questions about where the ‘Persona-modulated Prompt’ is placed.

---

> > > ### Author Response · Authors · 2023-11-17
> > >
> > > Thanks for your quick reply. We are happy to elaborate on this.
> > >
> > > First, we want to clarify a typo in our previous response (which is actually correct in the paper). Vicuna takes the persona-modulation prompt as system prompt. It is only Claude that takes it as user prompt. Claude does not support System prompts through the API, see their documentation [here](https://docs.anthropic.com/claude/reference/complete_post).
> > >
> > > Our prompts look as follow for all three models (they are all then formatted according to provider templates, but we use common formatting here for clarity). The assistant message after the persona modulation prompt is included to improve model compliance.
> > >
> > > **GPT-4 and Vicuna**
> > > ```
> > > SYSTEM: <persona modulation prompt>
> > >
> > > ASSISTANT: I understand and I will respond accordingly
> > >
> > > USER: <misuse_instruction>
> > >
> > > ASSISTANT:
> > > ```
> > >
> > > **Claude**
> > > ```
> > > USER: <persona modulation prompt>
> > >
> > > ASSISTANT: I understand and I will respond accordingly
> > >
> > > USER: <misuse_instruction>
> > >
> > > ASSISTANT:
> > > ```
> > >
> > > Let us know if this clarifies your question. We are happy to provide further details.

---

> > > > ### Author Response · Authors · 2023-11-22
> > > >
> > > > As the deadline approaches, please let us know if our response has adequately addressed your concerns, or if you have any remaining questions. We really appreciate your review.

---

### Official Review · Reviewer_pyxV · 2023-10-31

**Soundness:** 1 poor
**Presentation:** 2 fair
**Contribution:** 1 poor
**Rating:** 3
**Confidence:** 3

**Summary:**

This proposes to design a persona modulation as a black-box jailbreak to make the LLMs to take on specific personas. Then it can comply with the harmful instructions and generate responses for harmful topics.

**Strengths:**

This paper proposes a persona modulation to enable LLMs to follow harmful instructions and generate corresponding responses.

**Weaknesses:**

1. **No novelty:** It seems this paper only designs a persona prompt. I don't see any novelty at all.

2. **Experimental setting has some issues:** "The authors of the paper manually labeled 300 random completions." Only authors of this paper annotated the completions. It might have some bias. Besides, baseline only has the one without prompt, which seems not enough.

**Questions:**

1. What about steering the system to be not only one personas? How about two personas in consistent or contradictory status?

2. It seems only one third of the categories achieved a harmful completion rate over 50%. Does that mean the proposed approach is not generalizable enough?

**Details Of Ethics Concerns:**

The paper proposed a method to tell LLMs to follow harmful instructions, and the corresponding outputs may be harmful!

---

> ### Author Response · Authors · 2023-11-16
>
> Thank you for your feedback. We have written a [general response](https://openreview.net/forum?id=gYa9R2Pmp8&noteId=2gC4CNvHds) that can also help clarify some of the issues you raised.
>
> > No novelty
>
> We have clarified our contributions in the updated version of the paper. We paraphrase them here:
>
> * **Finding successful persona-modulation prompts**: Telling the model to behave like a persona is not enough since they refuse by default since past persona-modulation attacks (like DAN) have been patched. Our first contribution is finding ways to enable persona-modulation again.
> * **Automating prompt generation**: This is one of our main contributions. We show that prompt engineering might no longer be required as LLMs become more capable and can craft jailbreaks at scale. We use GPT-4 to generate all persona-modulation prompts, which has not been done before in the literature. This makes attacks very scalable and cheap.
> * **Finding yet another vulnerability in SOTA safeguards**: Raising awareness on limitations of existing safeguards is important to show why models should not be blindly trusted and motivate better safeguards.
>
> > Bias in manual annotations
>
> Thanks for bringing this up. Crowdsourcing is known to have many problems [1,2,3] and authors' labels are generally accepted as gold-standard because authors are better trained for the task. Several authors independently labeled several splits to try to address this potential issue.
>
> > What about steering the system to be not only one personas? How about two personas in consistent or contradictory status?
>
> Having the LLM respond in two personas is something that we tested and found that it hampers longer free flowing conversations limiting the amount of useful information we can get from the model.
>
> > Does that mean the proposed approach is not generalizable enough?
>
> We have addressed this in our general response. Most of the failure cases are due to artifacts in the automated pipeline that can be solved with human oversight (semi-automated attack). We included a new paragraph in the discussion adressing this issue. Also, in practice, an attacker could improve performance with approaches like  best-of-n-sampling.
>
> -----
>
> [1] Veselovsky, Veniamin, Manoel Horta Ribeiro, and Robert West. "Artificial Artificial Artificial Intelligence: Crowd Workers Widely Use Large Language Models for Text Production Tasks." arXiv preprint arXiv:2306.07899 (2023).
>
> [2] Peng, Andi, et al. "What you see is what you get? the impact of representation criteria on human bias in hiring." Proceedings of the AAAI Conference on Human Computation and Crowdsourcing. Vol. 7. No. 1. 2019.
>
> [3] Michael Chmielewski and Sarah C Kucker. 2020. An mturk crisis? shifts in data quality and the impact on study results. Social Psychological and Personality Science, 11(4):464–473.

---

> > ### Author Response · Authors · 2023-11-20
> >
> > As the rebuttal phase deadline approaches, we wanted to check if you have any concerns that our comments and updated version have not addressed. We want to make sure all major concerns are fixed for the meta-review.

---

> > > ### Comment · Reviewer_pyxV · 2023-11-22
> > > **Reply to the rebuttal**
> > >
> > > I appreciate the author's effort in providing the responses. After carefully reading the author's rebuttal, I prefer to keep my current scores.

---

> > > > ### Author Response · Authors · 2023-11-22
> > > >
> > > > Thank you for your feedback. We would like to ask what exactly was not fixed/addressed in our updated version and rebuttal. Do you still consider our contributions not novel enough and/or authors' annotations inappropriate as a baseline?

---

> > > > > ### Comment · Reviewer_pyxV · 2023-12-01
> > > > > **Further response**
> > > > >
> > > > > Thanks for the explanation. I have no concerns about the annotation problem now. However, I still think the novelty of this paper is limited. Specifically, the technique novelty, including the machine learning algorithm and model architecture, doesn't have enough novelty to be accepted by a top machine learning conference.

---

### Official Review · Reviewer_DNY8 · 2023-11-01

**Soundness:** 3 good
**Presentation:** 3 good
**Contribution:** 3 good
**Rating:** 5
**Confidence:** 4

**Summary:**

The paper investigates persona modulation as a method to jailbreak language models at scale. It shows persona modulation can automate the creation of prompts that steer models to assume harmful personas, increasing harmful completions. This jailbreak transfers between models and is semi-automated to maximize harm while reducing manual effort.

**Strengths:**

- The paper presents a novel and thorough study of persona modulation to jailbreak LLMs
- Evaluating the attack against many different harm categories and several state-of-the-art models demonstrates the approach can generalize to different scenarios
- The methodology is clearly explained and the experiments are well-designed overall.

**Weaknesses:**

- Although the attack achieves a very high one-off attack rate, the attack pattern remains fixed and obvious, which can potentially be identified by some adversarial classifier. It is unclear whether the proposed attack would still be effective if the target model is fine-tuned on the attack prompt with safe answers provided.
- Factors that make some personas/prompts more effective than others are not analyzed.
- Concrete mitigation strategies are not discussed in detail.

**Questions:**

Was any analysis done on what factors make some personas and prompts more effective than others in the automated workflow?
What are some ways model providers could mitigate this attack specifically going forward? If some kind of mitigation is performed, can you estimate how the attack rate will change?

---

> ### Author Response · Authors · 2023-11-16
>
> Thank you for your feedback. We have written a [general response](https://openreview.net/forum?id=gYa9R2Pmp8&noteId=2gC4CNvHds) that can also help clarify some of the issues you raised.
>
> > The attack pattern remains fixed and obvious
>
> Our intuition is that taking on certain personas is a desirable feature that LLM providers want to have in their models to enable functionalities like the OpenAI Marketplace. Thus, defending against it might not be trivial to solve.
>
> Also note that every single persona-modulation prompt is different from each other and requires personalization (this is why automation is important). Defending against a wide set of prompts is not trivial.
>
> The fact that “attacks are fixable” is generally true for the entire jailbreaking literature that focuses on finding current vulnerabilities to inform future safety measures. With this work we want to keep showing new failures modes that prove a more general concern: **fixing specific attacks from the literature does not bring safety by default because new attacks (like ours) can be devised.**
>
> Finally, we circumvent SOTA safety measures in GPT-4 and Claude 2 showing this attack vector is currently not easy to detect nor obvious.
>
> > Factors that make some personas/prompts more effective than others are not analyzed.
>
> Thanks for bringing this up. **We included a discussion paragraph** on “limitations with generating instructions”. Since we only use API access to models that either comply with our harmful instructions or provide very general evasive responses, it is hard to disentangle what exactly makes a prompt more effective than others. We, however, identified that the issues are often intermediate artifacts in the automated steps. That is why including a human in the loop is effective and we use human oversight to get responses for prompts shown to be fixed in the GPT-4 system card (see Appendix E).
>
> > Concrete mitigation strategies are not discussed in detail.
>
> Please, see our general response. We believe that **fixing specific attack vectors does not provide a robust solution to the underlying inherent vulnerability in LLMs**. Additionally, it is hard to devise new mitigations since we do not know what measures are already implemented by LLM providers. This is also in line with most recent work on jailbreaking LLMs (e.g. [1]).
>
> -----
>
> [1] Wei, A., Haghtalab, N., & Steinhardt, J. (2023). Jailbroken: How does llm safety training fail?. arXiv preprint arXiv:2307.02483.

---

> > ### Author Response · Authors · 2023-11-20
> >
> > As the rebuttal phase deadline approaches, we wanted to check if you have any concerns that our comments and updated version have not addressed. We want to make sure all major concerns are fixed for the meta-review.

---

### Author Response · Authors · 2023-11-16
**General comments (PDF was updated)**

We would like to thank all the reviewers for their time and constructive feedback. **We have updated the submission PDF addressing some of the issues raised by reviewers and improving clarity.**

-----

## Evaluation of our method and baselines

**Evaluating harmfulness of the model generations is a hard problem** (see for example [1]). We believe that GPT-4 provides a good proxy for classification. In fact, we manually label 300 examples and compare them with GPT-4 judgements. In general, our GPT-4 classifier tends to be more conservative and provides a robust lower bound for performance (see Appendix A for details on detector evaluation). We included a paragraph in the discussion on this issue.

-----

## Effectiveness and mitigations for the attack

Some reviewers suggested this attack should be easy to fix. We argue against this point because GPT-4 and Claude 2 already include extensive efforts to ensure safety that can be circumvented with our attack. In any case, the main point we want to convey with our paper is that **models are vulnerable to adversarial prompts no matter how much effort is put into creating better safeguards**. Some of the attacks presented previously in the literature are now fixed, but our attack is still successful, and we expect this trend to hold true in the future. We argue that defending against a specific attack is not enough and that research should disclose as many concerning attack vectors as possible.

For these same reasons, in line with previous work on jailbreaks, we do not explore specific mitigation strategies since we believe that they will not address the broader safety limitations of LLMs. Additionally, it is not possible to contribute to safeguards of SOTA models since they are proprietary and unknown to the wider community.

-----

## Limited effectiveness for some categories

We found that most limitations are related to artifacts in the automated pipeline. Our assistant model (GPT-4) is sometimes not good at crafting questions about sensitive topics. This is why semi-automated attacks can improve performance substantially. We have included a paragraph in the discussion addressing “limitations with generating instructions”.

-----

## Disclosing specific prompts

We decided **not to** disclose them publicly to prevent misuse of SOTA models by anyone with very little effort. We are, however, sharing the details with trusted researchers upon request. We included a “broader impact” paragraph discussing this decision in the updated PDF.

-----

[1] Zou, Andy, et al. "Universal and transferable adversarial attacks on aligned language models." arXiv preprint arXiv:2307.15043 (2023).

---

### Meta-Review · Area_Chair_b8Ts · 2023-12-05

**Metareview:**

The paper proposes an effective approach for jailbreaking Large Language Models (LLMs) using persona modulation. It demonstrates that persona modulation can automate the creation of prompts leading LLMs to assume harmful personas, thereby increasing harmful completions. The method is tested on several state-of-the-art models including GPT-4. The 42.5% attack success rate for GPT-4 is impressive. And the method is straightforward and easy to understand. Despite all the strengths, the evaluation of the method can be improved. The authors could provide more insightful experiments on defense, effective factors to help us understand the mechanisms of this approach better, before the submission or during the rebuttal.

In summary, the work holds a lot of promise but could benefit from some more careful studies.

**Justification For Why Not Higher Score:**

The decision to reject hinges on a few key points:

Limited novelty as perceived by some reviewers.
Concerns about the fixed and identifiable nature of the attack pattern.
Lack of detailed analysis on what makes certain prompts more effective.
Lack of more comprehensive evaluations.

**Justification For Why Not Lower Score:**

N/A

---

### Decision · Program_Chairs · 2024-01-16

Reject